# Bridging the gap: Enhancing HIV care pathways for young key populations in Chad

Esias Bedingar[1,2]*, Sabrina Ebengho[1], Ferdinan Paningar[3], Ngarossorang Bedingar[4], Eric Mbaidoum[5], Naortangar Ngaradoum[5], Aisha K. Yousafzai[1]

1 Department of Global Health and Population, Harvard T.H. Chan School of Public Health, Boston, Massachusetts, United States of America, 2 Alma, Centre de Recherche en Systèmes de Santé, Porte 107 Chagoua, N'Djamena, Chad, 3 Bucofore, Quartier Béguinage, Rue Joseph Brahim Seid, N'Djamena, Chad, 4 Croix Bleue Tchadienne, Porte N128, arrondissement, Chagoua, N'Djamena, Chad, 5 Réseau National des Personnes Vivants avec le VIH, N'Djamena, Chad

* esias.bedingar@gmail.com

## Abstract

Young key populations—sex workers and men who have sex with men (MSM)—face significant barriers to accessing HIV care in Chad due to stigma, discrimination, and socio-economic challenges. Although legal protections exist, gaps in enforcement continue to undermine care efforts. This study explored care pathways for young key populations in Chad to identify their specific challenges and propose targeted strategies to strengthen the HIV care continuum. Conducted in April 2025 in N'Djamena Chad, the qualitative study involved ten in-depth interviews with sex workers and MSM, aged 15–24 years, recruited through snowball sampling. Framework analysis revealed key themes across the HIV care continuum, specifically in testing, linkage to care, and retention in antiretroviral therapy (ART). Three major themes and 13 sub-themes emerged: (1) HIV testing and diagnosis, (2) linkage to care and ART initiation, and (3) retention in care and adherence to ART. Stigma—both externalized and internalized—was a significant barrier at every stage. Fear of disclosure, discrimination in healthcare settings, and financial constraints further hindered care engagement. Conversely, community-based awareness programs in faith-based institutions, and peer support networks were crucial in facilitating access to services. Addressing these challenges requires a comprehensive, multi-level approach that includes stigma-reduction training for healthcare workers, enforcement of anti-discrimination laws, targeted financial aid mechanisms, and integrated psychosocial counseling. Strengthening community-based interventions and peer-led outreach can further enhance engagement and retention, thereby improving health outcomes and reducing new infections among young key populations in Chad while aligning with global HIV targets.

## Introduction

Key populations—including, sex workers, men who have sex with men (MSM), transgender people, people who inject drugs, and people in prisons and other enclosed settings—experience significant inequalities in accessing HIV prevention, treatment, and care globally [1, 2]. These groups are disproportionately affected by the HIV epidemic, facing heightened risks of

**Data availability statement:** The data underlying this article cannot be shared publicly due to ethical restrictions (Harvard T.H. Chan School of Public Health's IRB [#IRB23-1743]). When applying for ethical approval, we did not specify that the data would be made publicly available in a repository. As part of the written and verbal consent, we assured participants that all data would be confidential and access to the recordings would be restricted to the research team. We did specify that "some of their words" may be used in reporting the findings of the study (which we have done within the manuscript as non-identifiable quotes), however, to make all raw data publicly available will be a serious breach to the rights of ethical of participants given did not consent to this. You can reach out to Leslie Howes (lhowes@hsph.harvard.edu or orarc@hsph.harvard.edu) for queries regarding data access for legitimate research.

**Funding:** This work was supported by the Tessa Jowell Fellowship for Doctoral Research, Harvard University (to BE) and the Prime Minister's Office, Chad (to BE). The funders had no role in study design, data collection and analysis, decision to publish, or preparation of the manuscript.

**Competing interests:** The authors have declared that no competing interests exist.

infection and numerous barriers to care, primarily due to stigma, discrimination, and legal obstacles [1–3]. In 2022, more than half (55%) of all new infections globally occurred among key populations and their sexual partners, while in sub-Saharan Africa (SSA), this group accounted for 25% of new infections [4, 5]. Structural inequalities such as poverty, gender disparities, and fragile healthcare systems exacerbate HIV transmission among key populations, limiting access to services [6, 7]. In Chad, punitive laws further restrict access to HIV care by criminalizing same-sex relationships, sex work, and drug use, leading to increased police harassment, social exclusion, and violence [8–13]. For instance, Chad's 2017 Penal Code (Article 354) criminalizes same-sex relationships, while sex work remains illegal under Articles 335 and 336, exposing sex workers to exploitation and arrests [14, 15]. These legal and social barriers, compounded by discriminatory healthcare practices, discourage individuals from seeking HIV testing, treatment, and preventive measures such as pre-exposure prophylaxis (PrEP) [16, 17].

The situation is even more challenging for young key populations (ages 15–24 years), who face additional age-specific barriers, including legal dependency, lack of comprehensive sexual health education, and limited youth-friendly healthcare services [3,9,18,19]. Globally, youth accounted for 28% of new HIV infections and 6% of all AIDS-related deaths in 2023, with 44% of new infections occurring among adolescent girls and young women [20–22]. Compared to older adults, youth exhibit lower ART adherence, greater treatment attrition, and poorer outcomes, often due to social stigma surrounding adolescent sexuality and HIV status [23–28]. Chad's youth-driven HIV epidemic reflects these global trends, with 26.3% of new HIV infections in 2022 occurring among 15–24-year-olds, with young women accounting for 19.5% of new infections—four times the rate of young men [29, 30]. While national data on young key populations remain limited, existing estimates indicate high HIV prevalence and disparities in ART coverage and condom use among MSM, sex workers, and people who inject drugs [30].

Recognizing these challenges, Chad's National Strategic AIDS Response Plan and the National Health and Development Strategy for Adolescents and Young People prioritize HIV prevention and treatment for vulnerable youth, particularly adolescent girls [30, 31]. Despite recent legal reforms, protections for young key populations remain weak, with the continued criminalization of MSM, sex work, and drug use limiting access to HIV services [30–34]. The 2019 UNICEF Multiple Indicator Cluster Survey reported widespread discriminatory attitudes, with 53.3% of women and 44.7% of men holding negative perceptions of people living with HIV [35]. The 2021 Behavioral Surveillance Survey Report found that 97% of MSM avoided healthcare due to stigma and discrimination [36]. In response, 0 *on the Prevention, Care, and Control of HIV/AIDS*, integrating a comprehensive prevention package targeted at sex workers, MSM, and prisoners [33,37]. The revised National Strategic AIDS Response Plan aims to reduce new HIV infections by 36% by the end of 2025, with a focus on young key populations [30]. However, implementation challenges persist, and many young key populations remain unable to access HIV care due to structural barriers and weak enforcement of legal protections.

HIV services in Chad are provided through public health centers, hospitals, and community clinics, following a standard continuum that includes voluntary HIV testing, linkage to care, ART initiation, and long-term retention [38]. However, young people face multiple disruptions in this pathway due to stigma, financial barriers, and legal concerns [38]. Many prefer alternative care-seeking options, such as pharmacies, traditional healers, and informal peer networks, to avoid discrimination in formal healthcare settings [38]. Young key populations are particularly vulnerable to delayed diagnosis and treatment interruptions, often resulting from fears of arrest, exposure, or mistreatment by providers. Strengthening youth-friendly,

stigma-free, and confidential HIV services is critical to improving engagement and retention in care among young key populations [38].

Understanding the barriers, facilitators, and structural challenges across the HIV care continuum for young key populations is critical for improving health outcomes. However, there is a major gap in the literature on how young key populations in Chad navigate HIV services, what specific challenges they face, and what facilitators help them stay in care. This study aimed to (1) explore the experiences of young key populations in accessing HIV care in Chad, (2) identify barriers and facilitators across the care continuum, and (3) propose strategies to strengthen linkage to care, adherence, and long-term retention. By addressing both structural and individual-level challenges, this research contributes to efforts to bridge the HIV care gap and align Chad's response with global HIV targets, such as the UNAIDS 95-95-95 goals [39]. Implementing youth-centered, stigma-free interventions is crucial to ensuring that young key populations receive equitable and effective HIV care.

## Methods

### Study design and sampling

This study is a secondary data analysis utilizing qualitative data from a parent study that employed a grounded theory design [40, 41]. The original data were gathered to understand how youth perceive and utilize sexual and reproductive health (SRH) and HIV services in Chad. Given the marginalized nature of these populations, a snowball sampling method was employed to identify eligible participants while ensuring safety and privacy. Participants were recruited through the Chadian National Network of Associations of People Living with HIV (RNTAP+), which also serves key populations. Eligible criteria included: (1) being aged 15–24 years; (2) identifying as MSM or a sex worker, and (3) willingness to discuss experiences with HIV testing, linkage to care, ART initiation, and retention. The sample included 10 participants (n=5 MSM; n=5 sex workers), ensuring representation across key populations at heightened risk of poor HIV care outcomes. Although qualitative studies do not aim for generalizability, the sample size was deemed sufficient for achieving data saturation, meaning no new themes emerged in the final interviews. Informed consent ensured voluntary participation and ethical compliance.

### Data collection

Ten in-depth interviews (IDIs) were conducted on April 30, 2024, by four locally trained interviewers with expertise in working with key populations and sensitive health topics [41]. Interviewers underwent specialized training to ensure cultural competency, confidentiality, and ethical sensitivity when engaging with marginalized youth. To ensure participant comfort and trust, female interviewers conducted interviews with female sex workers, and male interviewers with MSM. Interviews were conducted in French and Arabic, lasting approximately 60 minutes, in private settings to protect confidentiality. A semi-structured topic guide explored barriers and facilitators across the HIV care continuum (S1 Text). All interviews were audio-recorded with consent, transcribed verbatim, imported into ATLAS.ti version 22, and analyzed in the original languages (French and Arabic) before translation into English to preserve contextual meaning [42, 43]. A rigorous quality assurance process, including transcription cross-checking, peer verification, and double-coding using ATLAS.ti version 22, ensured data accuracy and reliability [43].

### Data analysis

Data were analyzed using a framework analysis approach to systematically examine HIV care pathways for young key populations [44, 45]. The analysis followed five structured

steps: familiarization, identifying a thematic framework, indexing, charting, and mapping and interpretation [45–47]. During familiarization, two authors (BE and FP) independently reviewed transcripts to identify emerging themes [46]. A coding framework was then developed based on the study's objectives and key stages of the HIV care continuum, including (1) HIV testing and diagnosis, (2) linkage to care and ART initiation, and (3) retention in care and adherence to ART. Using ATLAS.ti version 22, data were systematically coded to ensure consistency and reliability. In the charting phase, themes were organized into matrices to facilitate cross-case comparison and synthesis (S1 Fig). The final stage of analysis involved mapping and interpretation, where patterns were identified to develop a comprehensive understanding of barriers and facilitators across the HIV care continuum. To enhance rigor, intercoder reliability was ensured through double-coding of a subset of transcripts, with discrepancies resolved through team discussions. Triangulation was achieved by comparing participant narratives with existing literature and consulting local experts to validate findings. These steps ensured that the analysis captured both the structural and individual-level challenges impacting HIV care engagement among young key populations in Chad.

## Research trustworthiness

The study adhered to criteria of credibility, transferability, dependability, and confirmability [48]. Credibility was enhanced through triangulation and member checking, sharing initial findings with participants for validation. Transferability was facilitated by detailed contextual descriptions. Dependability was ensured by maintaining an audit trail documenting the research process and analysis steps. Confirmability was achieved through reflexivity, acknowledging potential biases, and ensuring findings were derived directly from the data.

## Reflexivity statement

The diverse research team engaged in continuous self-reflection to identify and mitigate biases. As a native of Chad, the first author brought essential local insights and cultural sensitivity, while collaboration with RNTAP+ ensured the study's grounding in local reality. Locally trained interviewers, fluent in local languages and cultural nuances, captured authentic data. Regular team meetings and a reflexive journal helped maintain awareness of biases, discussions across the study team ensured findings were based on participants' realities rather than researchers' preconceptions.

## Ethical considerations

Ethical considerations were critical due to the sensitivity of the topic and vulnerability of the youth participants. Ethical approval was obtained from the Harvard T.H. Chan School of Public Health's IRB (#IRB23–1743) and the National Committee on Bioethics of 0). Informed consent was obtained from participants aged 18 and above, while assent and guardian consent were secured from minors under 18. To protect confidentiality, pseudonyms were used, and identifying details were removed. Data were securely stored and accessible only to the research team. Measures to minimize distress included same-gender interviewers for sensitive discussions and providing participants with counseling resources. Additional information regarding the ethical, cultural, and scientific considerations specific to inclusivity in global research is included in the Supporting Information (S2 Text).

## Patient and public involvement statement

Participants were not involved in the design, or conduct, or reporting, or dissemination plans of this research. As mentioned above, credibility during the study was enhanced through

triangulation and member checking, sharing initial findings with participants for validation. At the end of the study, we shared findings from the study with members of the community who were not interviewed, individuals who are service providers or experts on the issue within the community, as well as stakeholders.

## Results

A total of 10 young key population participants (n=5 MSM; n=5 female sex workers) were interviewed through IDIs, with a mean age of 20 years. The MSM group had a mean age of 18 years (SD = 3 years), while the female sex worker group had a slightly higher mean age of 22 years (SD = 2 years). The analysis revealed distinct barriers and facilitators affecting the HIV care continuum for sex workers and MSM. This stratification provided nuanced insights into the specific challenges and supports encountered by these groups. A total of three themes and 13 sub-themes were identified: (1) HIV testing and diagnosis; (2) linkage to care and ART initiation; (3) retention in care and adherence to ART (Table 1).

**Table 1. Themes identified through focus group discussions with participants.**

| Subthemes | Definition |
| --- | --- |
| *Theme 1: HIV testing and diagnosis* | |
| Self-stigma | Internalized negative beliefs and feelings about oneself due to fear of being HIV positive, leading to reluctance in seeking testing. |
| Stigma | Negative societal attitudes, beliefs, and stereotypes toward individuals based on their association with HIV or sexual orientation, leading to feelings of shame and self-isolation. |
| Discrimination | Unjust treatment or prejudiced actions by healthcare providers or society due to an individual's HIV status or sexual orientation, resulting in avoidance of HIV testing. |
| Limited access to HIV-related information | Limited awareness or knowledge about where and how to access HIV testing services, often due to inadequate outreach or healthcare communication. |
| Community-based awareness programs | Initiatives conducted within communities to educate and encourage individuals to get tested for HIV, often reducing fear and stigma. |
| Supportive environment | Safe, non-judgmental settings that encourage HIV testing by reducing the fear of stigma and discrimination. |
| *Theme 2: Linkage to care and ART initiation* | |
| Discrimination within healthcare facilities | Poor treatment and prejudice from healthcare providers due to patient's HIV status or sexual orientation, leading to reluctance in seeking care. |
| Lack of tailored healthcare services | Absence of tailored healthcare services that address the unique needs of sex workers and MSM, leading to difficulties in accessing appropriate care. |
| Faith-based organizations | Healthcare facilities associated with religious organizations that provide a trusted and supportive environment for initiating ART. |
| Peer support networks | Assistance and encouragement provided by peers within the same community, which plays a crucial role in linking individuals to care and initiating ART. |
| *Theme 3: Retention in care and adherence to ART* | |
| Fear of disclosure | Persistent anxiety about others discovering one's HIV status, particularly in healthcare settings where confidentiality might not be guaranteed. |
| Financial constraints | The inability to afford consistent care or medications, leading to interruptions in treatment. |
| Confidentiality and trust in healthcare providers | Assurance from healthcare providers that patient information will be kept confidentiality, which improves retention in care and adherence to ART. |
| Community-based and home-based support services | Regular follow-ups and support provided by healthcare workers or peer educators within the community or at home to ensure adherence to ART. |

## Theme 1: HIV testing and diagnosis

### Subtheme 1: Barriers to testing and diagnosis

Participants cited stigma, self-stigma, lack of information, and fear of disclosure as major barriers to HIV testing. Stigma emerged as a pervasive barrier to HIV testing, with both external (societal) and internal (self-stigma) factors discouraging individuals from seeking care. Many participants described facing negative societal attitudes due to their association with HIV, sex work, or same-sex relationships, which led to shame, isolation, and avoidance of testing services. A sex worker shared, *"People look at us like we are already sick, even before we test. It makes it hard to go to the hospital"* (PS2, 20 years old). MSM participants similarly expressed fear of being publicly labeled based on assumptions about their HIV status, reinforcing low engagement with testing services. One participant explained, *"It's to find out my HIV status and because it's a place of public care and everyone goes there, and it allows me to hide in the crowd within my community, so as not to be indexed"* (PR1, 16 years old), highlighting how MSM often strategically choose larger, more general health-care settings to blend in, reducing the risk of being singled out or facing stigma within their communities.

Self-stigma was also a major psychological barrier, as individuals internalized societal prejudices, leading to fear and denial about their health status. One sex worker expressed, *"Fear and doubt have been the main causes for not seeking SRH and HIV services"* (PS2, 20 years old). The anticipation of a positive diagnosis often triggered intense anxiety, making the prospect of testing overwhelming. This internal struggle was compounded by the awareness that a positive result could lead to further social alienation and discrimination, not only from the public but also from within their immediate community. Another sex worker added, *"Even if I want to know my status, the thought of being HIV-positive terrifies me. I would rather not know"* (PS4, 24 years old).

Beyond stigma, MSM and sex workers expressed fear of discrimination from healthcare providers, leading them to avoid testing altogether. One participant shared, *"Discrimination and shame are the reasons why we do not seek HIV testing services"* (PR3, 23 years old), highlighting how stigma within healthcare settings discourages individuals from engaging with HIV care. Another participant described the judgmental attitudes of providers, stating, *"It's hard to get tested because we fear being judged and treated differently"* (PS1, 22 years old). This fear was further reinforced by negative experiences healthcare staff, as one MSM explained, *"Some nurses won't even touch us. You can see it in their faces—they think we deserve this"* (PR1, 16 years old). The avoidance of HIV services due to discrimination creates a vicious cycle in which individuals remain undiagnosed and untreated, further exacerbating their vulnerability to HIV-related complications. As another participant put it, *"Even when I try to ask, they make it seem like I'm wasting their time"* (PR3, 23 years old), illustrating how provider neglect and bias discourage key populations from seeking essential health services.

Access to accurate and reliable information was essential for encouraging HIV testing, yet many in the MSM community faced significant challenges in obtaining this information. The lack of targeted outreach and communication meant that individuals may not know where or how to get tested, which delayed or prevented from accessing services. An MSM highlighted this issue, stating, *"No one gives us information about SRH, but our doctors, when we approach them for treatment, they give to us. The same is true for HIV care"* (PR2, 15 years old). This reliance on healthcare providers as the sole source of information placed a heavy burden on individuals to seek out care proactively, which was difficult when faced with stigma and discrimination. Another MSM echoed this concern, noting, *"We don't have access to enough*

*information; we are often in the dark about where to go or who to ask questions"* (PR5, 15 years old). The absence of widespread, community-based information dissemination further exacerbated the problem, leaving many without the knowledge they needed to protect their health. A third MSM underscored the impact of this information gap, stating, *"Most of us find out about testing options only through word of mouth, and even then, it's not reliable"* (PR3, 23 years old). The lack of accessible information sources contributed to confusion and uncertainty, making it harder for individuals to navigate the healthcare system and access necessary HIV testing services.

Finally, fear of involuntary disclosure was a widespread concern, particularly among sex workers, who worried that a positive HIV test result would not remain confidential. Many participants feared breaches of privacy in healthcare settings, where staff could share their status with family members or within the community. One sex worker expressed this anxiety, stating, *"The worst part is not even knowing my status—it's knowing that someone else might know before I'm ready"* (PS1, 22 years old). For MSM, the risk of being outed due to an HIV test was also a significant deterrent. A participant shared, *"If I go to get tested, people might assume things about me, and that's dangerous"* (PR4, 21 years old). The double stigma of HIV and sexual orientation placed MSM in a particularly vulnerable position, leading some to delay or completely avoid testing to protect their social standing.

### Subtheme 2: Facilitators to testing and diagnosis

Community-based awareness programs played an important role in reducing fear and stigma around HIV testing, particularly for MSM. These programs, often conducted within community settings, offered peer-led education and non-judgmental support in a non-threatening environment, helping to alleviate the fear of testing. An MSM remarked, *"It used to be so hard for me because I thought that if I was positive, I was going to die. However, since after the awareness it was in a community setting, I overcame the stress"* (PR4, 21 years old). This indicated how community-driven initiatives can effectively address the psychological barriers that prevent individuals from seeking testing. By normalizing the conversation around HIV and providing peer support, these programs created a safer space for individuals to consider testing without the fear of judgment. Another MSM noted the importance of these programs in building confidence, stating, *"Awareness-raising, referral of healthcare providers with a pass, follow-up, withdrawal of results, referral for the management of ARVs"* (PR2, 15 years old). This structured approach to awareness and support ensured that individuals are not only informed about the importance of testing but are also guided through the process in a supportive and confidential manner.

Supportive healthcare environments, particularly private and faith-based institutions, emerged as critical facilitators of testing. These settings were perceived as more confidential, respectful, and better organized than public clinics, providing a safer space for individuals to access health services. An MSM noted, *"The service is easy for me to access. The amenities are not the same because private and religious institutions are better organized than the public ones"* (PR3, 23 years old). This preference for non-public institutions reflected the importance of an environment that prioritizes confidentiality and respect, reducing the fear of stigma and discrimination. A sex worker emphasized the value of supportive settings, stating, *"Private, denominational, and specialized institutions offer quality services in contrast to public institutions"* (PS3, 22 years old). The perceived quality and discretion of these institutions made them a more attractive option for those who might otherwise avoid testing due to fear of exposure. Supportive environments thus played a crucial role in facilitating access to HIV testing by providing a space where individuals feel safe, respected, and cared for.

## Theme 2: Linkage to HIV care and ART initiation

### Subtheme 1: Barriers to linkage to care and ART initiation

Discrimination within healthcare settings discouraged many participants from linking to care after an HIV diagnosis. Negative experiences with healthcare providers, who may exhibit prejudice based on a patient's HIV status or sexual orientation, deterred individuals from returning for care after an HIV diagnosis. An MSM described the impact of such discrimination, stating, *"It's the problem of discrimination and poor reception of healthcare professionals that make us decide not to seek care services"* (PR4, 21 years old). This highlighted how discriminatory attitudes within healthcare settings can lead to a breakdown in trust, causing patients to disengage from care. A sex worker expressed similar concerns, noting, *"We are dismissed when we seek information; even for SRH and HIV care, providers are reluctant to help us, as if it's pointless because they assume we're already beyond help"* (PS2, 20 years old). This sentiment reflects the deep-rooted stigma that persisted within healthcare systems, where individuals felt judged and marginalized, making them reluctant to pursue necessary treatment.

The lack of tailored healthcare providers for key populations presented a significant obstacle to their effective linkage to care and initiation to ART. Public healthcare facilities often did not cater to their unique needs, resulting in a gap in service provision that deterred them from seeking care. An MSM stated, *"There are no specific structures for our community"* (PR1, 16 years old). This lack of specialized services meant that MSM were forced to navigate a healthcare system that was not designed to address their specific risks and needs, leading to delays in care and treatment. Another MSM highlighted the importance of improving services for this population, noting, *"Specialized HIV care services need to make an effort to support our community and train their health workers to better serve our community"* (PR2, 15 years old). The absence of targeted support structures and trained personnel further exacerbated the challenges faced by sex workers and MSM in accessing and adhering to HIV care.

### Subtheme 2: Enablers to linkage to care and ART initiation

Faith-based organizations offered a critical pathway for sex workers and MSM to link to care and initiate ART, providing a trusted and supportive environment that contrasted with the often-hostile public healthcare system. Many MSM turned to these institutions for their perceived confidentiality, compassion, and respect for patient privacy. An MSM shared their positive experience, stating, *"I did my first [HIV test] at a faith-based institution because I knew I would be treated with respect and my privacy would be protected"* (PR4, 21 years old). This sense of safety and trust was critical, as it encouraged individuals to take the first step in managing their health. A sex worker highlighted the supportive environment, stating, *"The staff at the faith-based clinic didn't just treat me – they listened to me, which made all the difference"* (PS1, 22 years old). This supportive and non-judgmental care fostered trust, making it easier for individuals to commit to their treatment. As another sex worker noted, *"In a world where we are often judged, the faith-based clinic was the only place I felt truly cared for"* (PS2, 20 years old). This profound sense of care and acceptance within faith-based institutions not only helped in initiating ART but also strengthened adherence by providing a continuous support system rooted in respect and understanding.

Peer support was an essential component linking key populations to HIV care and ensuring the initiation of ART. The shared experiences and understanding within peer groups provide a foundation of trust and solidarity, making it easier for individuals to navigate the often-intimidating healthcare system. An MSM emphasized the importance of this support, saying, *"Having someone who's been through it makes a big difference – they understand the fears and can guide you on what to do next"* (PR1, 16 years old). The presence of peers who

have successfully engaged with HIV care can significantly reduce the barriers to treatment by offering practical advice and emotional encouragement. Another MSM described how peer support helped him overcome his initial fears, stating, *"I was really scared at first, but seeing others manage their treatment gave me the confidence to start mine"* (PR5, 15 years old). This support from peers helped individuals feel less isolated and more confident in taking the necessary steps toward managing their health. A third MSM reflected on the role of peer educators, saying, *"Screening, convincing participants to get tested to find out their serology at the site; in case of positive results, go to the health service, provide HIV care in specialized HIV care structures"* (PR2, 15 years old). These quotes illustrated how peer support networks not only provided critical information but also offered guidance and encouragement throughout the care process.

## Theme 3: Retention in HIV care and adherence to ART

### Subtheme 1: Barriers to retention in care and adherence to ART

Fear of disclosure remained a significant barrier to retention in care and adherence to ART among sex workers and MSM. The anxiety that their HIV status might be revealed without their consent, particularly by healthcare providers, often prevented individuals from continuing their treatment. A sex worker reflected on this issue, noting, *"Sex workers are afraid to continue to seek healthcare services because there is stigma in health services and confidentiality is not guaranteed"* (PS3, 22 years old). Moreover, the potential social and economic repercussions of such disclosure were severe, as an MSM emphasized, *"It's not easy to seek SRH and HIV care because we are a minority that is discriminated against and rejected a lot by society"* (PR3, 23 years old). This statement highlights how the fear of being "outed" extended beyond the clinic walls, affecting every aspect of their lives. The overwhelming dread of having their status exposed, whether intentionally or accidentally, often led sex workers and MSM to avoid healthcare settings, even when they desperately needed ongoing care. A sex worker elaborated, *"I've seen how people treat those who are known to have HIV; it's like your life is over"* (PS5, 20 years old). This reflects the deep-seated fear that seeking care could lead to further marginalization, not just from healthcare providers, but from society as a whole.

Financial constraints were a significant barrier to retention in care and adherence to ART for key populations. The costs associated with regular medical visits, transportation, and purchasing medications were cited as prohibitive, leading individuals to forgo care or ration their treatment. An MSM highlighted the financial difficulties they face, stating, *"The work of personal and emotional care is difficult and so is financially"* (PR3, 23 years old). This reflects the broader challenges of maintaining adherence to ART in the face of economic hardship, where the choice between healthcare and other basic needs can result in inconsistent treatment. Another MSM explained, *"I have to choose between buying food or paying for transport to the clinic"* (PR5, 15 years old). Additionally, the lack of financial support exacerbated these challenges, as expressed by a sex worker, *"We need more help; it's hard to keep up with the costs, and no one seems to care"* (PS4, 24 years old). This sense of being overlooked or unsupported by the healthcare system contributed to a cycle of poor adherence and deteriorating health. The financial burden was not just about the direct costs of care but also the indirect costs, such as taking time off work or the stress of managing limited resources.

### Subtheme 2: Enablers to retention in care and adherence to ART

Confidentiality and trust were vital components for ensuring retention in care and adherence to ART among key populations. The assurance that their HIV status and personal information will be kept confidential by healthcare providers was crucial for encouraging ongoing

engagement with care. A sex worker emphasized the importance of confidentiality, stating, *"Confidentiality and trust […]"* (PS2, 20 years old) when asked what factors were crucial for returning for viral load follow-up. This trust was built on the understanding that their privacy will be respected, which significantly influenced their willingness to return for regular care. Another sex worker shared, *"I only go to places where I know they won't share my information with anyone"* (PS4, 24 years old). This highlights the critical role that trust plays in healthcare decisions, as sex workers were acutely aware of the risks associated with potential breaches of confidentiality. The impact of trust – or the lack thereof – on their engagement with care was further underscored by another participant who noted, *"In some clinics, I'm afraid to even say why I'm there because I don't trust them to keep it private"* (PS1, 22 years old). On the other hand, when trust is established, it can create a supportive environment that fosters adherence to ART. As another sex worker described, *"When I trust the healthcare worker, I'm more likely to follow their advice and come back for my check-ups"* (PS4, 24 years old).

Community and home-based support played an essential role in retaining MSM in HIV care and ensuring adherence to ART. These personalized support systems, often provided by community actors within the community or at home helped MSM to stay engaged with their treatment. An MSM highlighted the value of this support, stating, *"I am consistently followed-up; they ensure that my medications are taken properly"* (PR2, 15 years old). Another participant underscored the impact of this personalized care, saying *"Having someone check in on me regularly made all the difference. It reminded me that I wasn't alone in this"* (PR3, 23 years old). This regular, compassionate engagement helped to address the unique challenges faced by MSM, particularly in environments where stigma and discrimination are prevalent. The close-knit support provided by community and home-based initiatives fostered a sense of belonging and trust, making it easier for individuals to follow through with their treatment plans. As another MSM noted, *"Knowing that someone cares and is there to help makes it easier to stick with the medication"* (PR1, 16 years old).

Across themes, we looked to see whether there were any distinct patterns based on characteristics of the participants, such as by gender, but did not find any key differences. Overall, there was a shared experience of these common barriers and enablers. However, it is worth noting that overall enablers were less frequently highlighted by participants compared to barriers (S1 Fig). While barriers across the HIV care continuum were widely discussed, enablers were often mentioned with less consistency and emphasis. This trend was observed across both groups – MSM and sex workers – suggesting that participants were more likely to focus on challenges rather than facilitating factors.

## Discussion

This study aimed to provides critical insights into the barriers and facilitators affecting the HIV care continuum among young key populations in Chad, specifically young sex workers and MSM aged 15–24 years. The findings highlight the profound impact of stigma, discrimination, and socio-economic barriers on accessing and adhering to HIV care. Additionally, supportive environments, peer networks, and community-based interventions emerged as crucial facilitators in improving engagement in HIV services.

### Barriers to HIV care

Our findings reaffirmed the pervasive impact of stigma and discrimination in limiting HIV testing, linkage to care, and retention in ART, echoing existing research in SSA. Fear of negative attitudes from healthcare providers was a significant barrier, leading many young key populations to avoid HIV services altogether. This aligns with previous studies

demonstrating how healthcare stigma discourages HIV testing and increased delayed treatment initiation among MSM and sex workers [49, 50]. Self-stigma was also a major barrier, as young key populations internalized societal prejudices, contributing to fear of a positive diagnosis and reluctance to seek care. These findings are consistent with prior research showing that self-stigma can lead to psychological distress and disengagement from HIV services [51]. Financial constraints were another significant challenge, particularly for retention in care and ART adherence. Given the young age of participants, many lacked financial independence, making difficult to cover transportation costs, clinic fees, and time away from work. This supports findings by [52], which highlight economic barriers as a major factor in HIV care disengagement [52]. The financial constraints faced by young key populations in this study may be largely attributable to their youth and the likelihood that many of them are not yet employed [52]. This lack of financial independence can exacerbate difficulties in covering both direct and indirect costs, such as paying for transportation to healthcare facilities [51, 52]. Similar findings have been reported, which highlight how the intersection of youth and unemployment increases vulnerability to financial barriers in accessing healthcare [53, 54].

## Facilitators of HIV care

Despite these barriers, several facilitators emerged as promising pathways for improving HIV care engagement among young key populations. Faith-based organizations played a complex role. Many participants cited religious health facilities as safe spaces where they received compassionate, non-discriminatory care, aligning with studies showing that faith-based organizations often provide essential care, support, and advocacy for people living with HIV, often leveraging their extensive networks and trust within communities [55, 56, 57]. While our study found that faith-based organizations provided a trusted and compassionate environment for HIV testing and care, this finding differs from much of the existing literature. Studies such as Lariat et al. (2024) and [58] suggest that faith-based organizations may inadvertently perpetuate stigma and discrimination and acknowledge the need for greater inclusion of young key populations [3, 58]. Peer support networks were also identified as vital in facilitating HIV care. The role of peer support in improving health outcomes has been well-documented, with studies like those by [59] and [60] demonstrating how peer networks provide essential emotional and practical support, helping to overcome the barriers posed by stigma and discrimination. Our findings add to this body of knowledge by highlighting the specific impact of peer support within the context of young key populations in Chad, emphasizing the importance of these networks in enhancing engagement in HIV care [59, 60]. Community and home-based support emerged as another critical facilitator in our study, particularly for retaining MSM in HIV care. This finding aligns with recent literature that emphasizes the effectiveness of community-based interventions in improving ART adherence and retention in care. For instance, a study by Decroo et al. (2017) in Mozambique provided compelling evidence of the impact of community-based ART delivery models on improving patient outcomes, including higher rates of adherence and retention [61]. Decroo and colleagues implemented a model where ART was distributed directly within communities compared to the traditional facility-based models, significantly reducing the need for frequent visits to healthcare facilities [61]. This approach not only minimized the logistical and financial burdens on patients such as travel costs and time away from work, but also helped mitigate the stigma associated with visiting HIV clinics [61]. These findings resonate with the broader literature, which highlights the effectiveness of peer-led adherence clubs and home visits in sustaining ART adherence among marginalized groups [62, 63]. These studies underscore the importance of integrating community and home-based support into HIV care

programs to address the unique needs of MSMs and other key populations, particularly in contexts where traditional healthcare settings may not be fully accessible and welcoming.

## Implications for policy and practice

The findings from this study underscore the urgent need for comprehensive, multi-level interventions to address the persistent barriers young key populations face in accessing and adhering to HIV care in Chad. Stigma and discrimination within healthcare settings remain major obstacles that deter individuals from seeking testing and remaining in care. Despite the existence of legal frameworks, such as *Law No. 019/PR/2007 on Prevention, Care, and Control of HIV/AIDS*, these laws lack strong enforcement mechanisms and fail to explicitly address the needs of young MSM and sex workers. To create a more inclusive healthcare environment, health system reforms should include provider training programs focused on non-discriminatory, youth-friendly HIV care. Establishing anti-stigma policies in health facilities, such as anonymous feedback mechanisms and legal accountability for discrimination, can further increase trust and engagement with healthcare services [64]. Additionally, civil society organizations and human rights advocacy groups must be involved in monitoring the implementation of these legal protections, ensuring that young key populations are truly included in HIV policies rather than remaining an overlooked subpopulation [65].

The criminalization of same-sex relationships and sex work in Chad remains a significant structural barrier that discourages young MSM and sex workers from accessing health services. Article 354 of Chad's Penal Code criminalizes same-sex relationships, and Articles 335 and 336 fail to legally recognize sex work, subjecting individuals to arbitrary arrests and social marginalization [14,15,33,37]. Given the global evidence that punitive legal environments increase HIV vulnerability among key populations, it is crucial for Chad to adopt evidence-based legal and policy reforms [66–68]. While full decriminalization may face strong sociopolitical resistance, targeted legal and policy reforms can mitigate the impact of criminalization on HIV outcomes [69, 70]. One effective approach is the implementation of non-enforcement policies for health-seeking individuals. Countries like South Africa and Kenya have introduced health sanctuaries, where law enforcement refrains from making arrests in designated healthcare facilities [69]. Chad could adopt a similar model by establishing legally protected safe zones in clinics, ensuring that young key populations can seek healthcare without fear of legal consequences. Additionally, healthcare providers should receive clear guidelines that protect patient confidentiality, preventing them from reporting key populations to law enforcement. Another critical step is integrating legal aid services into the national HIV response. In Uruguay, a National Council for HIV/AIDS Prevention provides legal assistance to marginalized populations, helping them navigate discrimination and rights violations [69]. Chad can replicate this by partnering with human rights organizations to offer free legal aid clinics for young MSM and sex workers. Establishing third-party reporting mechanisms would allow victims of discrimination or violence to seek justice without exposing themselves to further harm. Training community-based paralegals from within key populations can further empower individuals to understand and exercise their legal rights. Many countries with similar legal restrictions have successfully introduced harm reduction policies and inclusive HIV frameworks even in the absence of full legal decriminalization [69–72]. Portugal, for example, decriminalized drug use and sex work, redirecting individuals into health and social support programs instead of the criminal justice system [69]. Chad could pilot alternative sentencing programs, where individuals arrested under current penal codes are referred to community service or HIV prevention programs instead of facing incarceration. Establishing harm reduction centers, where sex workers and MSM can access HIV prevention tools, counseling, and legal support, would help reduce their

vulnerability while maintaining public health priorities. Chad could leverage regional human rights mechanisms and international HIV commitments (e.g., UNAIDS Fast-Track Strategy) to advocate for greater legal protections for young key populations while maintaining cultural sensitivities.

Expanding financial support mechanisms is another important area for policy intervention. As the study reveals, financial constraints, including the costs of transportation, medical visits, and medications, significantly hinder the ability of young key populations to remain in care and adhere to ART. Implementing financial incentives to encourage HIV prevention behaviors among young key populations in Chad has shown promise in various contexts. Studies in low- and middle-income countries have demonstrated that conditional cash transfers and fixed monetary incentives can effectively increase HIV testing rates and reduce risky behaviors [73, 74]. While specific financial incentive programs for young key populations in Chad are not well-documented, the government has implemented initiatives such as the Prevention of Mother-to-Child Transmission of HIV (PMTCT) Programme in 2005, indicating a willingness to adopt innovative HIV preventive strategies [75]. However, the sustainability of financial incentives remains a key challenge. Long-term reliance on cash incentives is often financially unfeasible, given Chad's resource constraints and the risk of diminishing returns. To create a more sustainable impact, incentives should be complemented with income-generating activities (IGAs), which have proven successful in other settings [73–76]. Economic strengthening initiatives targeting vulnerable populations have been shown to improve retention in HIV care and adherence to ART. For instance, Uganda's Suubi + Adherence program combined savings-led economic empowerment with ART adherence support, resulting in improved viral suppression rates among adolescents living with HIV [76]. Similarly, microfinance, vocational training, and transportation assistance have been linked to higher engagement in HIV care and treatment [73–76]. For Chad, a tailored economic strengthening program for young key populations, particularly MSM and sex workers, could integrate vocational training, micro-enterprises, and financial literacy workshops to provide sustainable alternatives to high-risk work environments that increase their vulnerability to HIV. Microfinance initiatives offering small business loans could empower sex workers to transition into alternative employment, reducing reliance on sex work while improving financial security and access to healthcare. Additionally, social protection schemes, such as conditional cash transfers tied to healthcare visits or adherence milestones, could help alleviate financial barriers to HIV treatment while ensuring long-term sustainability. To be effective, these interventions must be implemented alongside legal reforms that protect young key populations from discrimination, ensuring a supportive policy environment where economic empowerment programs can thrive. By integrating financial incentives with sustainable economic strengthening strategies, Chad can enhance HIV prevention, improve treatment adherence, and foster long-term resilience among young key populations.

Decentralized, community-based HIV service delivery is another key strategy for improving HIV outcomes among young key populations. Given the documented success of community ART models in sub-Saharan Africa, Chad should expand decentralized HIV services by strengthening community-based ART distribution, mobile HIV testing units, and peer-led adherence support programs [61,63,77]. These models have been shown to increase ART retention by reducing travel burdens, mitigating stigma, and offering tailored support from community health workers [61–63]. Home-based HIV care, including doorstep ART delivery and virtual adherence counseling, could further address access barriers for young key populations who fear being seen at health facilities. Additionally, peer-led adherence clubs could provide psychosocial support and structured ART follow-up, fostering long-term engagement with care. Expanding these community-driven interventions would ensure that HIV services

are more accessible, youth-friendly, and responsive to the needs of young key populations in Chad.

Finally, these findings suggest that a more holistic approach to HIV care is needed—one that not only addresses medical needs but also considers the social, economic, and psychological factors that influence health behaviors. The mental health burden associated with HIV-related stigma also requires explicit integration into HIV care strategies. Many young key populations experience psychological distress, depression, and anxiety, which further contributes to delayed testing, poor ART adherence, and social withdrawal. Integrating mental health professionals into HIV service delivery can improve treatment retention and overall well-being. Recent studies have demonstrated that psychosocial counseling in HIV care settings improves ART adherence, reduces viral load, and enhances quality of life among key populations [78–80]. Chad's HIV response should include mental health screening, confidential counseling, and referral systems within both facility-based and community-based HIV programs. Training healthcare providers and peer educators in trauma-informed care and stigma reduction strategies will be essential to ensuring a more supportive and non-judgmental care environment for young key populations.

Finally, multi-sectoral collaboration is necessary to ensure that HIV policies are comprehensive and sustainable. Addressing the barriers faced by young key populations requires partnerships between the health sector, legal institutions, community organizations, and international development agencies. Strengthening government-NGO collaboration can mobilize resources, build advocacy networks, and facilitate the implementation of inclusive HIV policies. International donors and technical partners should prioritize funding for youth-focused and key population-specific interventions. Additionally, fostering regional partnerships with neighboring countries that have made progress in key population inclusion could provide valuable lessons and policy models for Chad. By implementing these policy recommendations, Chad can move towards a more inclusive and effective HIV response, ensuring that young MSM and sex workers receive the necessary care and support to improve their health outcomes.

## Strengths and limitations of this study

This study provides critical insights into the barriers and facilitators affecting the HIV care continuum for young key populations in Chad, a group that is often marginalized and under-represented in research. A key strength of this study is its qualitative approach, which allows for an in-depth exploration of the lived experiences of young MSM and sex workers. By using a framework analysis, we captured nuanced perspectives that may not be easily detected in quantitative studies. Additionally, the study was conducted in partnership with local organizations, ensuring cultural sensitivity and enhancing trust between participants and researchers. Another strength lies in its practical policy implications, as the findings highlight actionable strategies to improve HIV service delivery, economic empowerment, and legal protections for young key populations.

Despite these strengths, several limitations must be acknowledged. First, the small sample size (n=10) may not fully represent the diverse experiences of all young MSM and sex workers in Chad. While data saturation was reached, the findings cannot be generalized to all young key populations in the country, particularly those in rural areas, where access to HIV services may be even more constrained. Second, the reliance on snowballing sampling may have introduced selection bias, as participants were likely drawn from the same networks, potentially limiting diversity in perspectives. Additionally, the study's cross-sectional nature prevents an understanding of how barriers and facilitators evolve over time, highlighting the need for longitudinal research to assess changes in engagement with HIV care. Another limitation is the focus on MSM and sex workers, excluding other key populations at high risk of HIV, such

as people who inject drugs and transgender individuals. Future research should expand the sample to include a broader range of young key populations, providing a more comprehensive understanding of HIV-related challenges. Lastly, while the study identifies economic barriers to HIV care, it does not quantitatively measure the impact of financial incentives or economic empowerment programs on treatment adherence. Future studies should evaluate the effectiveness of these interventions to inform evidence-based policies. Despite these limitations, this study fills a critical gap in knowledge by shedding light on the challenges faced by young key populations in Chad and providing policy-relevant recommendations to improve HIV service delivery, legal protections, and economic opportunities for this vulnerable group.

## Conclusion

This study adds to the existing literature by providing an understanding of the barriers and facilitators affecting the HIV care continuum for young key populations in Chad, particularly MSM and sex workers. Stigma, discrimination, financial constraints, and legal barriers hinder HIV testing, linkage to care, and ART adherence, while faith-based healthcare, peer support, and community-based interventions show promise in improving outcomes. To enhance HIV service uptake, Chad should implement harm reduction policies, such as non-enforcement zones in healthcare facilities, alternative sentencing, and legal aid for young key populations. Economic empowerment programs, including vocational training, microfinance, and conditional cash transfers linked to ART adherence, can provide sustainable alternatives to high-risk activities. Expanding community-based HIV services and mobile ART distribution will help overcome stigma-related barriers and improve retention in care. These strategies align with UNAIDS' 95-95-95 targets, ensuring broader inclusion of young key populations in Chad's HIV response. Future research should assess the impact of financial and structural interventions and develop scalable, young key populations-friendly healthcare models. By integrating evidence-based and culturally sensitive approaches, Chad can bridge existing gaps in HIV care, protect key populations, and contribute to global HIV/AIDS reduction efforts.

## Supporting information

**S1 COREQ Checklist. COnsolidated criteria for REporting Qualitative research Checklist.**
(PDF)

**S1 Text. A semi-structured topic guide for young key populations.**
(DOCX)

**S2 Text. PLOS' policy on inclusivity in global health.**
(DOCX)

**S1 Fig. HIV care continuum analysis matrix for young key populations.**
(DOCX)

**S1 Table. HIV statistics for key populations.**
(DOCX)

**S2 Table. HIV care continuum sample coding for young key populations.**
(DOCX)

## Author contributions

**Conceptualization:** Esias Bedingar, Sabrina Ebengho, Ferdinan Paningar, Ngarossorang Bedingar, Eric Mbaidoum, Naortangar Ngaradoum, Aisha Yousafzai.

**Data curation:** Esias Bedingar, Ferdinan Paningar, Ngarossorang Bedingar, Naortangar Ngaradoum.

**Formal analysis:** Esias Bedingar, Ferdinan Paningar, Ngarossorang Bedingar.

**Funding acquisition:** Esias Bedingar, Ngarossorang Bedingar.

**Investigation:** Esias Bedingar, Ferdinan Paningar, Ngarossorang Bedingar, Eric Mbaidoum.

**Methodology:** Esias Bedingar, Ferdinan Paningar, Ngarossorang Bedingar, Eric Mbaidoum, Aisha Yousafzai.

**Project administration:** Esias Bedingar, Ngarossorang Bedingar, Eric Mbaidoum, Naortangar Ngaradoum.

**Resources:** Naortangar Ngaradoum.

**Supervision:** Esias Bedingar.

**Validation:** Esias Bedingar.

**Writing – original draft:** Esias Bedingar.

**Writing – review & editing:** Esias Bedingar, Sabrina Ebengho, Aisha Yousafzai.

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
