## [Decision Letter · Decision Letter 0]

17 Jan 2025

PGPH-D-24-02129

Bridging the gap: Enhancing HIV Care Pathways for Young Key Populations in Chad

Dear Dr. Bedingar,

Thank you for submitting your manuscript to PLOS Global Public Health. After careful consideration, we feel that it has merit but does not fully meet PLOS Global Public Health’s publication criteria as it currently stands. Therefore, we invite you to submit a revised version of the manuscript that addresses the points raised during the review process.

The manuscript has been evaluated by four reviewers, and their comments are available below.

The reviewers have raised a number of major concerns. They request improvements to the reporting of methodological aspects of the study, expansion of the Introduction, and further discussion and consideration to the limitations of the study.

Could you please carefully revise the manuscript to address all comments raised?

We look forward to receiving your revised manuscript.

Kind regards,

Helen Howard

Staff Editor

Journal Requirements:

2. Please provide separate figure files in .tif or .eps format.

3. Please provide an Author Summary. This should appear in your manuscript between the Abstract (if applicable) and the Introduction, and should be 150–200 words long. The aim should be to make your findings accessible to a wide audience that includes both scientists and non-scientists. Sample summaries can be found on our website under Submission Guidelines:

https://journals.plos.org/globalpublichealth/s/submission-guidelines#loc-parts-of-a-submission.

Additional Editor Comments (if provided):

Reviewers' comments:

Reviewer's Responses to Questions

**Comments to the Author**

1. Does this manuscript meet PLOS Global Public Health’s publication criteria ? Is the manuscript technically sound, and do the data support the conclusions? The manuscript must describe methodologically and ethically rigorous research with conclusions that are appropriately drawn based on the data presented.

Reviewer #1: Partly

Reviewer #2: Yes

Reviewer #3: Yes

Reviewer #4: Yes

2. Has the statistical analysis been performed appropriately and rigorously?

Reviewer #1: No

Reviewer #2: Yes

Reviewer #3: Yes

Reviewer #4: Yes

3. Have the authors made all data underlying the findings in their manuscript fully available (please refer to the Data Availability Statement at the start of the manuscript PDF file)?

Reviewer #1: No

Reviewer #2: No

Reviewer #3: No

Reviewer #4: No

4. Is the manuscript presented in an intelligible fashion and written in standard English?

Reviewer #1: Yes

Reviewer #2: Yes

Reviewer #3: Yes

Reviewer #4: Yes

5. Review Comments to the Author

Reviewer #1: The paper, Bridging the Gap: Enhancing HIV Care Pathways for Young Key Populations in Chad, tackles a critical issue, but several weaknesses undermine its impact. While the subject matter is important, the study fails to fully explore the complexities of the HIV care continuum for young key populations, and methodological limitations make it difficult to generalize the findings.

Major Issues:

1. With only 10 participants, the study's findings lack statistical power and cannot be considered representative of the broader population. This weakens the claims about common barriers and facilitators across the care continuum.

2. Focusing solely on MSM and sex workers excludes other relevant key populations, limiting the study's ability to provide comprehensive insights into HIV care challenges.

Overemphasis on Barriers: The results disproportionately highlight barriers while offering only cursory discussion of facilitators. This unbalanced approach makes the findings seem more pessimistic without suggesting practical solutions.

3. The discussion on faith-based organizations is overly simplistic, failing to adequately address how these entities might simultaneously facilitate care and perpetuate stigma.

Minor Issues:

1. The introduction is verbose, reiterating points that could be streamlined for a more concise narrative.

2. Recommendations are vague, lacking specificity about how to implement the suggested changes within Chad's existing policy framework.

In conclusion, while the paper addresses an urgent issue, it falls short in depth and scope. Future research should broaden the sample, include diverse key populations, and aim for a more balanced analysis of barriers and facilitators to provide actionable insights.

Reviewer #2: All above information are true and in honesty to my comments.

1. The manuscript meet PLOS Global Public Health’s publication criteria. The manuscript is technically sound, and the data support the conclusions.

2. The statistical analysis been performed appropriately and rigorously

3. The authors have not made all data underlying the findings in their manuscript fully available

4. The manuscript is presented in an intelligible fashion and written in standard English

Reviewer #3: The article is well-written and meets the required standards for publication although, the raw data were not provided for ethical reason. The research question is relevant, and the methodology is appropriately applied.

However, I would like to raise a concern regarding the sample size. While the study’s findings are insightful, the small sample size may limit the generalizability of the results.

It would be beneficial to either increase the sample size in future studies or include a more detailed explanation of how the sample size was determined, along with its potential impact on the validity and reliability of the findings.

Despite this limitation, the article offers valuable contributions to HIV research and can still provide meaningful insights to readers and prospective researchers.

Reviewer #4: Article review

This is an important article that qualitatively explored the pathways, access and adherence to HIV care among young key populations in Chad. The article provided important contextual recommendations on how to improve HIV care to improve health outcomes for these vulnerable populations in Chad and contributing in the fight against HIV and AIDS.

Abstract: No comments

Introduction:

Line 46, In 2022, more than half of all new infections occurred among people aged 15-49 years from key populations and their sexual partners. It is essential to clarify whether this data pertains to global or regional statistics. Additionally, it is mentioned that this group accounts for 25% of new infections, which contradicts the earlier statement about more than half.

Line 48-50, The HIV epidemic in SSA is exacerbated by widespread inequalities, including poverty, gender disparity, and fragile healthcare systems, all of which contribute to high transmission rates and limited access to interventions. Does this statement apply to the general population or specifically to key and vulnerable populations? I would argue that the emphasis should be on the risks associated with the population of interest for this manuscript – key and vulnerable populations.

Line 51-52; Punitive laws and social stigma further marginalize key populations, significantly hindering their access to essential HIV services [8, 9]. Could the authors clarify and provide more context about specific laws and regulations that criminalize same-sex relationships, sex work, and drug use in Chad?

Line 58, These challenges are even more pronounced for youth, particularly those aged 15-24 years [3, 9, 15]. Are you referring to youth in general or young among key and vulnerable populations? I would suggest this paragraph to focus on issues with young people who are key populations and not youths in general. This focus started to appear in line 71.

Line 81. I am curious why the plans emphasis the importance of addressing the needs of young girls and not both, girls and boy.

Line 82, are these laws specific to chad?

Line 85, Could you please provide more details about the “significant human rights-related inequalities and barriers”? Does this imply that certain populations are being criminalized, or are you referring specifically to the criminalization of sex work, drug use, and same-sex behaviors?

In line 91, In response, the Law #19, authors have to make proper reference and naming of this specific law or act.

Lines 97-104, I recommend adding narration about the current pathways to care for the general population in Chad. Additionally, it would be helpful to include a commentary on how these pathways differ for young key populations. This will provide valuable context for understanding the background of the results.

Methods

Design and sampling

Line 112, Participants were recruited through our partner organization… Can you describe who is “you”, the institution of the first author?

Lines 112-117, Since this is a secondary data analysis, were there specific recruitment criteria established prior to data collection to ensure the inclusion of at-risk youth? Or did the authors select data from the at-risk population that was part of the larger sample of youth during the analysis?

Line 120: The sample size seems small, given that more than one at-risk youth group participated. Were ten participants adequate a sample for understanding the experiences of at-risk youth pathways to HIV care in N'Djamena?

Lines 126-127, "To maintain reliability and integrity of the analysis, we conducted the analysis in the original languages of the transcripts before translating the results into English." Could the authors clarify this point? Is the translation based on the final quotes included in the results, or was the entire results section originally written in another language and then translated? What is (are) the name (s) of the original language(s)?

Lines 127-128, A rigorous quality assurance process ensured high transcript quality, with cross-checks and reviews for accuracy and consistency. What tool did the authors use for quality assurance (QA)?

Lines 132-133, The primary aim was to understand the challenges and barriers faced across the HIV care continuum, from diagnosis to ART initiation, retention, and adherence. I think the authors need to be consistent with the main aim of the study as in lines 99-100 it reads “This study aimed to explore these pathways to enhance linkage from HIV diagnosis to ART initiation, retention in HIV care, and adherence to treatment.”

Lines 146-147, Authors need to expound more on how triangulation was done and in what ways validation was done with the participants. Credibility was enhanced through triangulation and member checking, sharing initial findings with participants for validation.

Line 163-164: Could you please clarify whether assent was obtained from the guardians of participants under 18? There might be some confusion; the text should indicate that informed consent was obtained from individuals aged 18 and older, while assent was secured from participants below the age of consent (18).

Results

Line 178; Ten young key population participants, evenly divided between males and females, were interviewed. Key informants’ interviews (KIIs) or in-depth interviews (IDIs)?

Line 191: Negative attitudes and judgment are connected to stigma but are not precisely the same. I recommend that the authors address each experience separately and include relevant quotes. Begin with stigma, addressing both projected and self-stigma, followed by discrimination. Then, discuss the lack of access to HIV care and treatment-related information, and finally, consider the fear of disclosing HIV serostatus. In table 1, make sure subthemes are self-explanatory for example access to information = access to HIV related information, stigma should have a separate definition from discrimination; stigma= refers to the negative attitudes, beliefs, and stereotypes that society holds toward a particular group or individual because of a perceived characteristic, condition, or behavior. It is a social process that devalues people and labels them as "different" or "less than." While Discrimination is the unfair treatment or unjust actions taken against individuals or groups based on characteristics such as race, gender, disability, sexual orientation, etc. It is the behavioral aspect of bias and prejudice.

Line 202: This quotation relates to barriers concerning sexual and reproductive health (SRH) and HIV services overall; however, this section specifically addresses issues associated with HIV testing and diagnosis. “Fear and doubt have been the main causes for not seeking SRH and HIV services”

Discussion

In the opening paragraph, it is important to emphasize that the focus is on the young, at-risk population, rather than the overall categories of sex workers and MSM.

Implications for policy and practice

Policy implications must be clearly distinguished from law enforcement. It should be explicitly stated which specific policies need to be changed or implemented. Similarly, specific laws that require modification or enforcement should be clearly identified.

How feasible is the recommendation to provide financial incentives to this specific population? Are there similar programs in place for other at-risk groups, such as adolescents and young girls in Chad? How sustainable will these incentives be? Additionally, is there a possibility of empowering this population through income-generating activities?

Limitations

It is excellent for the authors to acknowledge that a small sample was using in this study and call for studies with robust sample size.

6. PLOS authors have the option to publish the peer review history of their article (what does this mean? ). If published, this will include your full peer review and any attached files.

**Do you want your identity to be public for this peer review?** For information about this choice, including consent withdrawal, please see our Privacy Policy .

Reviewer #1: No

Reviewer #2: **Yes: ** Juliana Aggrey

Reviewer #3: **Yes: ** Abdulhafiz Arafah Akilu-Dindi

Reviewer #4: No

---

## [Editor Report · Decision Letter 1]

12 Mar 2025

Bridging the gap: Enhancing HIV Care Pathways for Young Key Populations in Chad

PGPH-D-24-02129R1

Dear Dr Bedingar,

We are pleased to inform you that your manuscript 'Bridging the gap: Enhancing HIV Care Pathways for Young Key Populations in Chad' has been provisionally accepted for publication in PLOS Global Public Health.

Best regards,

Guillaume Fontaine, PhD, RN

Academic Editor